# Mechanism Analysis of Vascular Calcification Based on Fluid Dynamics

**DOI:** 10.3390/diagnostics13162632

**Published:** 2023-08-09

**Authors:** Shuwan Xu, Feng Wang, Peibiao Mai, Yanren Peng, Xiaorong Shu, Ruqiong Nie, Huanji Zhang

**Affiliations:** 1Department of Cardiology, The Eighth Affiliated Hospital of Sun Yat-Sen University, Shenzhen 518033, China; xushw9@mail2.sysu.edu.cn (S.X.); wangfengmed@163.com (F.W.); maipb@mail2.sysu.edu.cn (P.M.); 2Department of Cardiology, Sun Yat-Sen Memorial Hospital of Sun Yat-Sen University, Guangzhou 510120, China; suoxinsuo@gmail.com (Y.P.); shuxiaorong1111@163.com (X.S.)

**Keywords:** vascular calcification, fluid dynamics, shear stress, hydrodynamic properties

## Abstract

Vascular calcification is the abnormal deposition of calcium phosphate complexes in blood vessels, which is regarded as the pathological basis of multiple cardiovascular diseases. The flowing blood exerts a frictional force called shear stress on the vascular wall. Blood vessels have different hydrodynamic properties due to discrepancies in geometric and mechanical properties. The disturbance of the blood flow in the bending area and the branch point of the arterial tree produces a shear stress lower than the physiological magnitude of the laminar shear stress, which can induce the occurrence of vascular calcification. Endothelial cells sense the fluid dynamics of blood and transmit electrical and chemical signals to the full-thickness of blood vessels. Through crosstalk with endothelial cells, smooth muscle cells trigger osteogenic transformation, involved in mediating vascular intima and media calcification. In addition, based on the detection of fluid dynamics parameters, emerging imaging technologies such as 4D Flow MRI and computational fluid dynamics have greatly improved the early diagnosis ability of cardiovascular diseases, showing extremely high clinical application prospects.

## 1. Introduction

Vascular calcification is the abnormal deposition of pathological mineral components in the vascular system, which is closely related to numerous cardiovascular diseases and is regarded as an important risk factor for adverse cardiovascular events [1]. Clinical and epidemiological data show that vascular calcification is a common pathological manifestation prevalent in patients with atherosclerosis, chronic kidney disease (CKD), hypertension, etc. [2,3,4]. Additionally, vascular calcification can be found in about 80% of vascular injuries and 90% of coronary artery diseases. Previous studies believed that vascular calcification was a passive process of calcium phosphate deposition, but subsequent studies confirmed that vascular calcification was an active and controllable process similar to osteogenesis. This process encompasses the activation of osteogenic signaling, the osteogenic transformation of vascular smooth muscle cells (VSMCs), and other effectors mediating abnormal vascular calcification [5,6].

Flowing blood can exert a frictional force called shear stress on the vessel wall, but its effect can vary in hydrodynamic properties depending on the geometry and location of the vascular tree [7]. In complex blood flow environments, such as arterial branches and vascular bends, the shear stress acting on the vascular wall is different from the physiological magnitude [8]. Through the signal transduction of endothelial cells (ECs) and VSMCs, mechanical signals are converted into electrical and chemical signals, and thus the “switch” of vascular calcification is turned on [9]. During the process of vascular calcification, ECs and VSMCs play crucial roles. They not only activate osteogenic signals but also engage in crosstalk, jointly mediating calcification. Given the significant influence of fluid mechanics on cardiovascular diseases, emerging imaging technologies such as 4D Flow MRI and computational fluid dynamics (CFD) have emerged, greatly enhancing the diagnostic and treatment capabilities of related diseases. This paper aims to elucidate the mechanisms of vascular calcification induced by changes in fluid mechanics and introduce novel imaging techniques based on fluid mechanics, providing valuable insights for the development of clinical diagnosis and treatment strategies for vascular calcification.

## 2. Overview of Vascular Calcification

Vascular calcification refers to the abnormal deposition of calcium phosphate complexes in blood vessels. While it is commonly observed during the normal aging process in humans, extensive research has increasingly identified vascular calcification as the pathological basis of numerous cardiovascular diseases, with close associations to major adverse cardiovascular events. Depending on the site of mineral deposits, vascular calcification can be categorized into intimal calcification and medial calcification, and can also manifest as valve calcification and calciphylaxis [10].

Intimal calcification is mainly involved in the progression of atherosclerosis. Calcification of the atherosclerotic intima begins with microcalcifications, characterized by a size less than 20 μm, originating primarily from the lipid pool and early necrotic core, but can also occur in the fibrous cap [11,12,13]. These microcalcifications can gradually transform into sheet calcifications, nodular calcifications, or plaque ossification observed in peripheral arteries [11]. This morphological evolution often coincides with the development of plaques. Microcalcifications are often identified in high-risk plaques. Many imaging methods point to the small size of calcification as a predictor of unstable plaque, which can significantly increase the local stress level of plaque and increase the risk of plaque rupture [14], while extensive calcification is associated with plaque stability [15]. Medial calcification is a chronic systemic vascular disease commonly seen in diabetes, CKD, aging, etc. [16]. The medial layer of blood vessels exhibits disseminated and progressive calcium phosphate deposits, leading to progressive petrification and decreased compliance of the vessel wall [17]. Medial calcification increases the incidence of cardiovascular complications [18]. It is a strong and independent risk factor for cardiovascular death and predicts the risk of future coronary events in patients with diabetes [17]. Different types of calcification are involved in the occurrence of adverse cardiovascular events. Scientifically elucidating the underlying mechanisms of vascular calcification holds pivotal clinical significance in the realms of diagnosing, treating, and preventing multiple cardiovascular diseases.

Previous studies have indicated that vascular calcification is not solely dependent on a high-phosphate and high-calcium environment, but also involves abnormal activation of bone-forming signals and disruption of calcification defense mechanisms, as well as an imbalance between pro- and anti-calcification factors [19,20]. Pro-calcification factors encompass various cytokines, inflammatory factors, and extracellular matrix proteins, among others. These factors promote the deposition and accumulation of calcium ions, leading to vascular wall calcification [6]. Inflammatory cytokines such as interleukin-1β (IL-1β), tumor necrosis factor-α (TNF-α), and interleukin-6 (IL-6) can activate VSMCs, inducing their transdifferentiation into osteogenic-like cells and the secretion of pro-calcification proteins [21,22]. The dysfunction of cellular organelles, including mitochondrial oxidative stress, autophagy defects, and endoplasmic reticulum stress, contributes to the dysregulation of calcium homeostasis. Moreover, various signaling pathways, such as Wnt/β-catenin, protein kinase B (PKB), NF-κB, and SGK1, can also trigger VSMC osteogenic transformation, leading to vascular calcification [23,24,25]. Moreover, the process of aging facilitates calcification by stimulating bone-inducing signals, inflammatory cytokines, oxidative stress, and related factors [26]. In pathological environments such as high phosphorus and chronic inflammation, VSMCs can also release extracellular vesicles containing high levels of calcification markers and a small amount of calcification inhibitors, and play a role in promoting calcification in the vascular microenvironment [27,28]. Furthermore, the accumulation of extracellular matrix proteins provides a platform for calcium ion deposition. On the other hand, anti-calcification factors involve various signaling pathways and molecular factors; disruptions or dysregulation of these signaling pathways may weaken the anti-calcification effects, leading to calcium accumulation in the vascular wall.

Thus, the development of vascular calcification is associated with the intricate interplay between pro-calcification and anti-calcification factors. Enhancement of pro-calcification factors and attenuation of anti-calcification factors can promote the deposition and accumulation of calcium ions, resulting in vascular calcification. Maintaining a balance between these two types of factors may hold significant implications for the prevention and treatment of vascular calcification.

## 3. Altered Shear Stress Mediating Vascular Calcification

The cardiovascular system is arranged in a tree form, consisting of the aorta to the capillary beds scattered in various end organs, and finally forms a closed loop of material transport [29]. Blood vessels at all levels are constantly exposed to the pulsatile pressure formed by myocardial contraction, but their various geometric and mechanical properties lead to different hydrodynamic properties [29].

The blood flow will produce a frictional force called wall shear stress (WSS) acting on the vessel wall, whose magnitude is determined by the blood flow velocity gradient on the surface of the vessel wall and affected by blood viscosity, which is crucial to the regulation of vessel function [30]. As the first barrier of circulating blood, ECs are constantly exposed to the WSS of blood flow and sense the hydrodynamic changes of blood, then transmitting them to the full vascular layer [2]. According to the hydrodynamic characteristics of different locations, WSS varies in the vascular tree: from 1 dyne/cm^2^ in the venous endothelium to 40 dyne/cm^2^ in the arterial vessels [7]. In addition, blood flow patterns also vary with different locations in the arterial system: the straight part of the arterial tree is dominated by laminar flow, while the blood flow in the curved area, branch points (such as aortic arch, bifurcation of iliac vessels, etc.) of the arterial tree is disturbed [8].

In the context of laminar flow, the WSS distribution within a vessel adheres to the classical parabolic profile, wherein the shear rate attains its minimum value at the central region of the vessel lumen and progressively augments as it approaches the vessel wall. [31]. The WSS exhibits typical physiological properties and imparts EC protection through its distinct hydrodynamic characteristics. These protective effects include the downregulation of inflammatory cytokines, adhesion molecules, and oxidative stress [32,33], increased production of nitric oxide and prostacyclin, and increased expression of vascular protective transcription factors kruppel-like factor 2 (KLF2) and NF-E2-related factor2 (Nrf2) [34,35]. The above effects are involved in maintaining the barrier function and antithrombotic surface of blood vessels, as well as regulating vascular tone. Relying on the pre-stretching properties of elastin and collagen in the arterial wall, arteries can maintain a dynamic balance between growth and remodeling under a certain range of blood pressure [36]. On the contrary, the blood flow at the curved area and the branch point of the arterial tree is different from the normal laminar flow due to disturbance, forming a flow-separation zone including flow reversal and occasional turbulence [8]. This different flow pattern has lower friction against the vessel wall than the physiological laminar shear stress and is highly oscillatory due to flow disturbances, as shown in Figure 1. Low WSS or highly oscillatory WSS can induce vascular inflammation [37]. Hyperproliferation and apoptosis of ECs increase the permeability of cholesterol-rich lipoproteins into arteries [38,39,40]. Recent studies have demonstrated that coronary segments in low-WSS regions have a higher incidence of thin-cap fibroatheroma of lipid-rich plaques, and show more superficial and punctate calcifications (Figure 2) [41]. Furthermore, WSS will also alter the characteristic EC alignment [42]. In the straight region of the artery, the ECs are flattened and aligned in the direction of the laminar flow of blood [43]. In the bifurcated or high-curvature vascular area, the blood flow is disturbed by the spatial characteristics to form turbulence and reverse flow. The ECs are arranged in a cobblestone-like manner to increase the volume [43]. However, intriguingly, despite the significantly lower WSS in veins compared to arteries, veins exhibit minimal calcification. This phenomenon may be attributed to the combined effects of several factors. Notably, veins lack elastic fibers and smooth muscle layers, and the absence of smooth muscle osteogenic transformation is considered a significant contributing factor. Furthermore, the lower venous pressure compared to arterial pressure may reduce the risk of calcium ion deposition and calcification. Additionally, the presence of venous valves, maintaining unidirectional blood flow and preventing blood reflux, could also play a role in preventing calcification. These factors likely contribute to the relatively lower incidence of calcification in veins despite the lower WSS values.

It is worth mentioning that the difference in diameter between parent and daughter vessels at the vascular bifurcation is one of the factors affecting vascular calcification and atherosclerosis. Murray’s law is a measure of the relationship between mother and child vessel diameters [44]. This law points out that the cube of the radius of the parent vessel at the optimum vessel bifurcation should be equal to the sum of the cubes of the radius of the daughter vessels. In this case, WSS remains constant [45]. In general, the branching vasculature of the mammalian circulatory system obeys Murray’s law, but deviations from this law can sometimes occur, resulting in regions of low WSS [46]. Schoenenberger et al. [47] showed that patients with a high Murray ratio (diameter ratio of parent vessel to daughter vessel) had a higher content of dense calcium in coronary plaques, and a lower content of fibrous tissue and fibrous adipose tissue. This confirms that deviations from Murray’s law are associated with high calcification near coronary bifurcations. In summary, based on Murray’s law, vascular bifurcations are common areas where calcification occurs due to WSS. Specific lesion sites of vascular calcification underscore the important role of hydrodynamic changes in the diagnosis and progression of cardiovascular diseases.

## 4. Molecular Mechanisms of Mechanical Stress-Mediated Vascular Calcification

### 4.1. Mechanisms of Endothelial Sensing of Mechanical Stress

Mechanosensitive structures and ion channels together form the structural basis for ECs to sense blood hydrodynamic changes. Subsequently, these mechanical signals are transduced into intracellular chemical signals to regulate changes in gene expression and cell behaviors [48]. Numerous prior investigations have consistently elucidated the potential mechanosensitive and responsive elements present on the endothelial surface. These encompass an array of cellular adhesion proteins (e.g., VE-Cadherin, PECAM-1), tyrosine kinase receptors (e.g., VEGF receptor 2), G protein-coupled receptors (GPCRs), and the endothelial glycocalyx (eGC), among others [7,9,49]. These structures jointly participate in the fine transmission of mechanical signals from extracellular to intracellular, making ECs a highly sensitive signaling hub (Figure 3).

The opening of mechanosensitive ion channels on the plasma membrane is the earliest cellular event following mechanical stimulation by fluid flow or membrane stretching [50]. Ion channel opening leading to an increase in intracellular Ca^2+^ is considered the first step in the endothelial response to mechanosignaling [51]. Ca^2+^ are known to stimulate the activation of endothelial nitric oxide synthase (eNOS) and intermediate conductance calcium-activated potassium channels (IKCa). This activation facilitates vasodilation through the release of nitric oxide (NO) mediated by eNOS and/or membrane hyperpolarization induced by the opening of IKCa channels [52]. Different ion channels vary in structure, ion selectivity, and gating properties. Among them, Piezo1 is a mechanosensitive nonselective cation channel found in ECs and vascular VSMCs. It is activated by shear stress induced by changes in local blood flow, as well as by cell membrane stretching during elevated blood pressure, resulting in a steady increase in intracellular Ca^2+^ concentration. Once Piezo1 is open, downstream Ca^2+^-dependent calpains (proteolytic enzymes) are activated, leading to actin cytoskeletal element adhesion turnover, proteolytic cleavage, and the responsive alignment of ECs to hydrodynamics [53,54]. Members of the transient receptor potential (TRP) family of non-selective cation channels also play a crucial role in hydrodynamically mediated vasoconstriction and relaxation. Among these channels, TRP Vanilloid 4 (TRPV4) in the endothelium and TRP Channel 6 (TRPC6) and TRP Melastatin 4 (TRPM4) in the smooth muscle are considered classic examples. TRPV4 channels demonstrate high Ca^2+^ permeability and distinct single-channel conductance properties [55]. In contrast, while the TRPC6 channel also exhibits Ca^2+^ permeability, its single-channel conductance is significantly lower [56]. Unlike most TRP family members, TRPM4 channels are permeable to Na^+^ and K^+^ but not to Ca^2+^. Their impact on Ca^2+^ in SMCs is indirectly generated through membrane potential depolarization and the subsequent activation of voltage-dependent calcium (Ca^2+^) channels (VDCCs) [57].

EC responses to hydrodynamic changes extend from acute adaptations of ion channel functionality to sustained modulation of gene expression patterns [58]. Bone morphogenetic proteins (BMPs), as well as other ligands, receptors, and regulators of the transforming growth factor β (TGFβ) family, regulate vascular and valve calcification by modulating the phenotypic plasticity of multipotent progenitor lineages associated with vessels or valves [59]. Rutkovskiy et al. [60] studied the expression pattern changes of procalcification genes by applying periodic stretch to ECs. The results suggest that ECs respond to mechanical stress by upregulating the expression of the pro-osteogenic factor BMP2 and other osteogenic factors in vascular mesenchymal cells, which may promote vascular calcification. BMP4 has pro-inflammatory effects on the endothelium, causing endothelial dysfunction, hypertension, and vascular calcification, and may play a role in atherogenesis. Csiszar et al. [61] used artery endothelial cells (CAECs) and cultured mesenteric arterioles of human and rat coronary to confirm that laminar shear stress and cAMP/PKA pathway are important negative regulators of BMP4 in vascular endothelium. Thus, mechanical stress under physiological conditions exerts an endothelial protective effect, protecting blood vessels from calcification and atherosclerosis. Butcher et al. [62] cultured monolayer porcine aortic endothelial cells (PAECs) and porcine aortic valve endothelial cells (PAVECs) under a steady-state shear stress of 20 dyne/cm^2^ for 48 h. Both types of cells expressed similar antioxidant and anti-inflammatory genes under shear stress, suggesting a protective role of stable shear stress against calcification.

### 4.2. Osteogenic Transformation of VSMCs during Calcification

Different from other smooth muscle cells, VSMC originates from mesenchymal stem cells, which endows it with phenotypic plasticity and the ability to differentiate into a specific single-lineage on proper culture medium [63]. During calcification, VSMCs undergo a phenotypic transition to osteogenic-like, characterized by loss of smooth muscle markers and upregulation of osteogenic markers (Figure 4) [64]. Compared with osteoblasts, there are only small discrete calcified areas in VSMCs of osteoblast-like phenotype, and the expression of early osteoblast markers runt-related transcription factor 2 (Runx2) and transcription factor 7 (Sp7) is still significantly lower than those of osteoblasts. Therefore, osteoblast-like VSMCs are more regarded as the excessive morphology between normal VSMCs and osteoblasts [64].

The phenotypic transition of VSMCs to an osteogenic-like phenotype plays a significant role in mediating the calcification process occurring in the intimal and medial layers of blood vessels. This transition is characterized by the upregulation of bone-related transcription factors, namely, Runx2, msh homeobox 2 (Msx2), and sex-determining region Y-box 9 (Sox9) [65]. Current studies have shown that the triggering of VSMC osteogenic transformation may be closely related to pathological factors, such as inflammation, mechanical stress, oxidative stress, aging, etc. In addition, the microcalcifications in the vessel wall may trigger the inflammatory response of the vessel, further accelerating the calcification process [66,67]. Calcified arterial media and atherosclerotic plaques express various regulatory bone formation and structural proteins. The canonical Wnt signaling cascade regulates osteogenic responses during vascular calcification. As one of the target genes of the Wnt cascade, Runx2 is the main transcription factor leading to the switch of osteoblast-like phenotype in VSMCs [68]. In response to different hydrodynamic features, the Wnt cascade induces vascular calcification in arterial tissue through Runx2, possibly with the cooperative involvement of matrix receptor integrins, intercellular receptor cadherins, and BMPs [69,70,71]. In the inflammatory microenvironment within the vascular wall, the activation of NADPH oxidase and the resultant elevation of hydrogen peroxide levels exert a stimulatory effect on the expression of Runx2. Subsequently, this increased expression of Runx2 serves as a pivotal trigger in facilitating the phenotypic transition of VSMCs towards an osteogenic-like phenotype [71]. Moreover, the inflammatory mediator tumour TNF-α can activate the expression of Msx2, which upregulates Runx2 and osterix through Wnt signaling, thereby driving the osteogenic transformation of VSMCs [65,72,73]. Relevant studies have shown that the receptor activator of nuclear factor-κB ligand (RANKL) can promote the calcification process by inducing the production of pro-osteoblastic cells in the vessel wall, and induces VSMC calcification through stimulating the expression of BMP4 [74]. Osteoprotegerin (OPG) also exists in the blood vessel wall, which can bind to RANKL and block its biological activity. Therefore, OPG knockout mice exhibit early and extensive vascular calcification [75]. It is worth mentioning that in a high-glucose environment, the AGE/RAGE signaling pathway severely affects cellular and systemic responses through PKC, p38 MAPK, fetuin-A, TGF-β, NF-κB, and ERK1/2 signaling pathways. This process increases BMPs and promotes the conversion of VSMCs to an osteoblast-like phenotype [76].

Matrix Gla protein (MGP) is highly expressed in VSMCs and chondrocytes, and its Gla residues have a high affinity for calcium ions, conferring potent inhibitory effects on calcification [77]. MGP may also be associated with the functional inhibition of BMP2 and BMP4, blocking calcium crystal deposition and shielding lesions from calcification [78]. Studies in MGP-deficient mice revealed disrupted medial elastic lamellae and calcification, leading to VSMC transdifferentiation into osteochondrocytic-like cells and loss of vascular elasticity [79]. Based on its carboxylation and/or phosphorylation status, MGP exists in different forms, but only fully carboxylated and phosphorylated MGP can exert inhibitory effects on calcification. In arterial media, where calcium deposition among VSMCs is prevalent, peripheral MGP is mainly present in an inactive, low-carboxylated form. Chronic kidney disease patients with arterial sclerosis often exhibit increased levels of inactive, non-phosphorylated MGP [80,81] and decreased serum concentrations of active, fully carboxylated MGP [82]. Therefore, achieving full activation of MGP holds promise as a potential approach for the treatment of vascular calcification.

Arterial tissue responds both structurally and chemically to various hydrodynamic properties to maintain its homeostasis. Physiologically, arteries experience constant levels of cyclic stress. However, when subjected to local low shear stress, vascular smooth muscle cells (VSMCs) have a tendency to differentiate into osteoblast-like cells within a disrupted homeostatic environment. This leads to the transformation of the arterial matrix into a bone-like matrix and promotes the formation of calcified plaques [83,84,85]. Nikolovski et al. [86] employed a 3-D engineered smooth muscle tissue model that expressed bone-related genes such as osteopontin, matrix gla protein, alkaline phosphatase, and the transcription factor CBFA1. The aim was to investigate the influence of the mechanical environment on bone gene expression and calcification. The results revealed that the engineered tissues exposed to cyclic strain exhibited a downregulation of bone-related genes, indicating a protective effect against calcification. In contrast, tissues subjected to no cyclic strain demonstrated increased calcium deposition. These findings suggest that in the absence of an appropriate mechanical environment, smooth muscle cells undergo a phenotypic shift towards an osteoblast-like pattern. Davenport et al. [87] exposed human aortic endothelial cells (HAECs) and human aortic smooth muscle cells (HASMCs) to physiological levels of hemodynamic (periodic) strain. The findings revealed that both cell types can produce OPG but not RANKL. Interestingly, the addition of exogenous RANKL prevents the “protective” upregulation of OPG mediated by cyclic strain at physiological levels.

In addition to the impact of shear stress under pulsatile blood flow, the mechanism underlying the transformation of VSMCs into an osteogenic phenotype in response to changes in the stiffness of the surrounding extracellular matrix is an area of significant interest. Matrix stiffness, a critical determinant of mesenchymal progenitor cell differentiation, exerts a profound influence on osteogenesis, with higher matrix stiffness promoting this process [88]. Cells possess the ability to sense changes in matrix stiffness through receptors such as integrins and DNA damage response (DDR) proteins [89]. This sensing triggers the transmission of resistance signals via the actin cytoskeleton, resulting in the activation of RhoA, a member of the Rho GTPase family. RhoA activation facilitates the formation of stress fibers by promoting the aggregation of actin and myosin [90]. In adipose stromal cells and fibroblasts, DDR1 responds to changes in matrix stiffness by interacting with the cytoskeletal protein nonmuscle myosin IIA (NMMIIA), even in the absence of β1 integrin [91]. Ngai et al. [92] conducted a study demonstrating that in VSMCs, the nuclear localization of Runx2 and the expression of osteochondrocytic markers exhibited an increase in a DDR1-dependent and matrix stiffness-dependent manner. Furthermore, the activity of RhoA and actomyosin contractility positively influenced the nuclear localization and expression of Runx2, as well as long-term calcification. During disease progression, DDR1 plays a role in sensing the stiffening of the matrix and regulating VSMC transdifferentiation into osteochondrocytic cells by transmitting contractile forces through the actin cytoskeleton.

### 4.3. Crosstalk between ECs and VSMCs

ECs and VSMCs within the arterial wall are spatially separated by an internal elastic lamina (IEL), which assumes a fenestrated structure facilitating the transport of water and small molecules [93]. The fenestrae present in the IEL enable direct contact between ECs and VSMCs, giving rise to myoendothelial junctions (MEJs) that establish a conduit for bidirectional signaling between these cell types [94,95,96].

The regulation of vascular tone and the response to blood-induced mechanical stimuli are inseparable from the synergistic effect of ECs and adjacent VSMCs. They not only co-express mechanosensitive ion channels, but also achieve the internal transmission of chemical signals relying on autocrine/paracrine mechanisms. ECs release NO after sensing shear stress, which diffuses to VSMCs and triggers vasodilation in a cGMP-dependent manner [97]. However, the reduction in NO is closely related to the increase in reactive oxygen species (ROS) levels in the vessel wall. ROS is produced by NAD(P)H oxidase, xanthine oxidase, or uncoupled eNOS [98]. It not only directly leads to the removal of NO, but also mediates the interruption of some signaling pathways related to NO generation. Additionally, vasoactive substances, including prostaglandins and endothelins, also act as regulatory substances, released in response to changes in hydrodynamics [99,100]. They mediate the increase or decrease in vessel diameter, which is called flow-mediated dilation (FMD). The reduction of NO production can lead to FMD injury, which is more common in diseases such as hypertension and hyperaldosteronism. In these diseases, blood vessels are considered endothelial dysfunction due to stagnation in a state of chronic vascular sclerosis [101].

In order to investigate the underlying mechanisms of interaction between ECs and VSMCs, in vitro co-culture perfusion systems have been widely employed [102]. These systems involve the cultivation of ECs and VSMCs on opposite sides of porous membranes coated with ECM, mimicking the structural arrangement of the IEL. Under static conditions, the co-culture of ECs with VSMCs induces a phenotypic shift characterized by enhanced cell proliferation and migration, along with a transition towards a proliferative and inflammatory phenotype. This phenotypic alteration is accompanied by a reduction in eNOS synthesis, upregulation of NF-κB signaling pathway, increased expression of adhesion molecules, and inflammatory mediators [103,104,105]. Exogenous application of laminar shear stress has been shown to alleviate EC dysfunction. It exerts its protective effects by reducing the expression of inflammatory genes such as IRAK, IKKg, IL6R, and CHUK. Additionally, it inhibits excessive EC proliferation and promotes EC health by inducing the expression of Sirt1 and down-regulating Cx40 [104,106]. Specifically, elevated shear stress levels (15 dyn/cm^2^) have been shown to increase NO production of EC and enhance the binding affinity of ET-1 to VSMCs. Conversely, under conditions of pathological low shear stress, ECs exhibit increased production of ROS through an AT1R-dependent pathway, resulting in vasoconstriction [107]. Interestingly, under static conditions, co-cultured VSMCs also exhibit migration and proliferation phenotypes, and laminar shear stress can inhibit the above cell phenotypes [108]. Conversely, low shear stress or turbulent flow can stimulate VSMC migration and proliferation by inducing ECs to produce FGF-4, VEGF-A, and PDGF-BB, which depend on the activation of CAV-1 or ERK in ECs [109]. The above studies indicated that ECs interact with VSMCs relying on paracrine mechanisms. ECs affect the function of VSMCs by secreting active substances such as NO, prostaglandin, ATP, growth factors, etc. However, the mechanism of VSMCs on ECs needs further study. Moreover, circulating leukocytes (monocytes, neutrophils, and lymphocytes) can also interact with ECs, independently of shear stress [110]. Laminar flow mediates contraction of pseudopodia in leukocytes, decreased activity of GPCRs, and decreased expression of the integrin CD18 in circulating leukocytes. This may inhibit the interaction of circulating leukocytes with ECs, resulting in functional effects on VSMCs to regulate vascular homeostasis and dysfunction [111].

Arteriovenous fistula (AVF) is the preferred vascular access for hemodialysis. Venous calcification rarely occurs, but in chronic hemodialysis patients, the incidence of AVF calcification is as high as 40% to 65% [112,113]. As an arterialized superficial vein segment formed after an anastomosis of an artery and a superficial vein, AVF exposes the venous ECs to hyperbiological shear stress [114]. As the AVF intima is shed due to repeated cannulation, the media is directly exposed to the fluid shear force of the blood and is more susceptible to chemical signals (phosphate molecules), which may contribute to the venous osteogenesis. Yang et al. [115] established a venous cell model of AVF using a fluid shear device combined with high phosphorus medium to simulate a uremic environment. The results showed that hydrodynamic stimulation had an additive effect on the upregulation of the osteogenic marker Runx2 in VSMCs at an early stage under shear stress and high phosphorus environment. The integrin β1-ERK1/2 signaling pathway contributes to the osteogenesis of venous SMCs under shear stress.

## 5. Blood Flow Imaging Technology Based on Fluid Dynamics

In recent years, technological advances in computer science and imaging have significantly propelled the field of cardiovascular imaging. Traditional cardiovascular imaging primarily focuses on depicting the geometric characteristics of the cardiovascular lumen and its changes during the cardiac cycle. However, blood flow imaging techniques, which are rooted in fluid dynamics, have expanded the scope of cardiovascular imaging. In addition to conventional vessel morphology, these techniques encompass the physical properties of the fluid, such as flow velocity and pressure distribution, at any given time. The laws governing blood flow are governed by mass and momentum conservation equations [116]. The mass conservation equation states that the sum of inflow and outflow rates in a small space is equal to zero. On the other hand, the momentum conservation equation describes the balance of forces acting on the fluid, including convective or “swirling” flow, pressure gradient, and friction due to viscosity. The mechanical stress information derived from blood flow is often indicative of potential adverse cardiovascular events. For instance, excessive WSS in coronary arteries can increase the likelihood of plaque rupture, whereas too-low WSS can lead to plaque progression [117]. Capturing the fluctuation or oscillation information of WSS is crucial for assessing the occurrence and progression of diseases such as vascular calcification or atherosclerosis. Flow energy loss (EL) is defined as the energy dissipation caused by turbulent flow in diseased conditions [116]. A significant increase in flow energy loss can not only indicate ventricular function overload but also serve as an effective predictor of heart failure occurrence [118,119]. Surgical interventions aimed at effectively reducing flow energy loss can lead to improved long-term cardiac function while reducing cardiac workload.

### 5.1. Four-Dimensional Flow MRI

A prominent emerging technology, 4D Flow MRI, introduces an innovative in vivo approach to assess fluid mechanics parameters. Technically, 4D Flow MRI builds upon phase-contrast magnetic resonance imaging (PC MRI) by incorporating three-directional flow velocity encoding and time-resolved capabilities [120]. It empowers the comprehensive measurement of fluid mechanics parameters within the heart and major vessels, providing full volumetric coverage throughout the entire cardiac cycle. The resulting data (3D + time + three velocity directions) facilitates the calculation of a multitude of derived fluid mechanics parameters, encompassing wall WSS, EL, pressure gradient, and pulse wave velocity, among others, significantly enhancing the evaluation of 3D fluid mechanics in vivo [121]. Furthermore, 4D Flow MRI excels in precision measurements of intricate blood flow patterns, exemplified by its ability to accurately assess false lumen flow in aortic dissections or entry/re-entry flow in cardiac valve diseases [122,123]. In addition, 4D flow MRI data are typically acquired during free breathing, utilizing retrospective electrocardiogram (ECG) gating with an end-expiratory navigation gate to synchronize the cross three-way blood flow coding gradient-echo pulse sequence [124]. This technique enables the retrospective evaluation of flow information across the entire vascular system from any image plane, as both anatomical and flow information are incorporated for every pixel in a 3D volume. Moreover, 4D flow MRI allows for time-resolved cine velocity acquisition. However, it is important to note that the time dimension in this context does not represent real time but is an effective average derived from multiple heart cycles. Consequently, any fluctuating or pulsatile changes in blood flow have a minimal impact. To visualize 4D flow MRI data, various techniques are employed, including streamline, isosurface, vector field, and volume rendering. Furthermore, the volumetric acquisition enhances the ability to accurately quantify complex cardiac blood flow [125]. With the rapid advancement of highly accelerated imaging and cutting-edge technologies, including artificial intelligence, 4D Flow MRI is expected to achieve higher speeds at a lower cost [126]. Given its ability to provide comprehensive spatiotemporal velocity vector data closely related to vascular pathophysiology, 4D Flow MRI has been applied in intracranial and cardiac vascular imaging. In the cardiovascular domain, 4D blood flow MRI is predominantly used for congenital heart diseases, but it also holds significant importance in other cardiovascular conditions, such as aortic aneurysms, aortic stenosis, pulmonary hypertension, and heart valve diseases [127].

However, there is still a lot of room for the development of 4D Flow MRI. Plenty of recent studies have explored the basis of it. 18F-sodium fluoride (18F-NaF) is a marker of calcification activity that identifies active regions of vascular calcification, the levels of which correlate with future disease progression and adverse events [128]. Minderhoud et al. [129] developed a software to explore the correlation between WSS and 18F-NaF PET uptake through precise co-registration of 4D flow MRI and 18F-NaF PET data. It can analyze the potential link between WSS and bicuspid aortic valve. However, 4D Flow MRI technology is slightly insufficient in time and space resolution, and the quantitative determination of fluid dynamics parameters takes a long time. Winter et al. [130] developed a set of post-processing algorithms based on a flexible reconstruction framework. It can greatly shorten the evaluation time of global pulse wave velocity and 3D-WSS, and significantly improve the analysis efficiency of in vivo measurements. Garrido-Oliver et al. [131] developed and tested a fully automatic aortic 4D flow MRI analysis process based on machine learning. Its automatic blood flow assessment results are in excellent agreement with manually measured in-plane and through-plane rotational flow descriptors, and axial and circumferential WSS, greatly improving the quantification efficiency of blood flow velocity.

### 5.2. Computational Fluid Dynamics

Different from the direct measurement data of 4D flow MRI, CFD calculates hydrodynamic parameters through computer simulation of a fluid model. This imaging technology can provide extremely high temporal and spatial resolution, and can be effectively applied to small blood vessels such as coronary arteries that cannot be detected by MRI [132]. Meanwhile, in some highly diseased flow with extremely high-velocity turbulent jets, the hydrodynamic parameters including WSS and EL can still maintain accuracy. For patients with coronary calcification, the diagnostic performance of coronary computed tomography angiography (CCTA) is often affected by calcification artifacts [133]. Compared with invasively measured fractional flow reserve (FFR), CFD model-based CT-FFR showed very high diagnostic performance in all lesions suspected of coronary artery disease (CAD), according to a Chinese multicenter study. In particular, the high specificity, sensitivity, and accuracy of CT-FFR, even in patients with calcifications, are significantly superior to previous CCTA assessments [134]. In addition, CFD modeling can also generate equations of fluid dynamics based on the geometry and fluid parameters of the patient’s vessels [116]. By combining “virtual surgery” with computer graphics simulations, CFD can guide optimal surgical decision making. However, CFD also has the disadvantage of being time-consuming. The high cost of modeling cardiovascular hemodynamics with available resources has also hindered the incorporation of CFD into clinical practice [135]. Acquiring accurate blood flow parameters in a low-cost and efficient manner is crucial for the diagnosis of cardiovascular diseases. AI, especially computer deep learning technology, makes it possible to achieve the above goals. Machine learning enables instantaneous or fast predictive outcomes. Su et al. [136] incorporated the blood flow WSS values of 2000 ideal coronary arteries calculated by CFD simulation into the machine learning model. By adopting multivariate linear regression, multi-layer perceptron, and convolutional neural network architectures, it is possible to directly generate WSS values within 1 s without using CFD. Through comparing the computational accuracy, the convolutional neural network outperforms other architectures, with a normalized mean absolute error of 2.5%. Lv et al. [137] designed a fast, end-to-end, pixel-wise AI-based platform, including an automatic segmentation platform for aortic segments and a time-averaged wall shear stress (TAWSS) computing platform. By incorporating the CFD data set to train the AI model, it can automatically estimate the value and distribution of the ascending aorta TAWSS, with an ideal clinical application prospect. Gharleghi et al. [138] used computer deep learning technology to predict the luminal WSS magnitude of coronary artery bifurcations throughout the cardiac cycle based on steady-state solution, vessel geometry, and additional global features. Compared with CFD-derived values, the model has a deviation of <5% and an average calculation time of <2 min, which ensures high-fidelity predictions while achieving fast calculations. Computational deep learning techniques have significantly reduced the computational cost of large-scale population-based studies involving coronary hemodynamic metrics, and may open the way for future clinical integration.

### 5.3. Other Imaging Techniques

Hemodynamic modeling based on three-dimensional quantitative coronary angiography (3D-QCA) has potential value in detecting obstructive lesions or lesions with a vulnerable phenotype. The model can reconstruct coronary anatomy and assess lesion severity in real time, thus providing another attractive option for WSS calculations [139]. The 3D-QCA-derived WSS correlates well with the WSS estimated by models reconstructed by intravascular imaging data [140]. Additionally, 3D-QCA-derived WSS can also identify vulnerable plaque and stratify cardiovascular risk in patients with a borderline negative FFR (FFR: 0.81–0.85) [141]. Assessment of WSS along plaques will help to better characterize vulnerable plaques, defining individualized risk of plaque rupture and stroke. Quantitative flow ratio (QFR) is a technique for calculating FFR based on 3D-QCA and CFD. Without pressure guide wire and hyperemia induction, it is superior to angiographic guidance in terms of the number of stents implanted, the dose of contrast agent applied, and the operation time. In a large-sample, multicenter, randomized, controlled clinical trial (FAVOR III China) conducted by Xu et al. [142], compared with angiography-guided percutaneous coronary intervention (PCI), QFR-guided PCI significantly improved the 1-year clinical endpoints of patients, with less myocardial infarction and ischemia-driven revascularization, showing a better prospect of broad clinical application of QFR. Conventional ultrasound imaging cannot accurately assess flow velocity in contact with the arterial wall. Ultrafast vector Doppler can calculate WSS by measuring the velocity vector of the entire 2D image. The setup for measuring WSS has been validated in vitro on a linear flow phantom by comparing measured values with in silico calculations [143]. The method provides a means of delineating the hemodynamic constraints of arterial plaques. The use of deep-learning-based ultrasound imaging for flow-vessel dynamics (DL-UFV) has the opportunity to improve the measurement accuracy of vascular properties and hemodynamics. Park et al. [144] designed an integrated neural network for super-resolved localization and vessel wall segmentation. It can be used to measure velocity field information of blood flow by combining with tissue motion estimation and flow measurement techniques. Through validating the evaluation results in mouse carotid arteries of different pathological types (aging and diabetes), DL-UFV is demonstrated to outperform conventional ultrasound flow and strain measurement techniques in measuring vessel stiffness and complex flow-vascular dynamics.

## 6. Conclusions and Perspective

Different levels of blood vessels exhibit distinct hydrodynamic properties. At the bending areas and branch points of the arterial tree, there are regions of blood flow reversal and turbulent flow separation. In such disturbed flow regions, the low WSS increases the vascular permeability of cholesterol-rich lipoproteins, disrupting the dynamic balance of arterial growth and remodeling under physiological conditions, and making it more susceptible to calcification. ECs utilize mechanosensitive structures such as cell adhesion proteins, GPCRs, and the endothelial glycocalyx to sense changes in blood flow dynamics. As the first barrier between the vascular wall and blood flow, the endothelium converts mechanical signals into electrical and chemical signals, thereby transmitting hydrodynamic changes throughout the vessel layer. The coordination between ECs and VSMCs is critical for regulating vascular tone and responding to blood-induced mechanical stress. These cells not only co-express mechanosensitive ion channels but also rely on autocrine and paracrine mechanisms for the internal transmission of chemical signals. Disturbances in blood flow patterns can trigger the transformation of VSMCs into an osteogenic-like phenotype, characterized by the upregulation of bone-related transcription factors, including Runx2, Msx2, and Sox9. These factors play crucial roles in mediating the calcification processes that occur in the intima and media of blood vessels. Understanding the pathways involved in calcification in both ECs and VSMCs is essential for unraveling the underlying mechanisms of vascular calcification and identifying potential targets for prevention and treatment. Additionally, advanced imaging technologies such as 4D Flow MRI and CFD are valuable tools for assessing physical parameters of blood flow. They enable the early detection of abnormal mechanical stress information during adverse cardiovascular events, thereby improving the early diagnosis of related diseases.

In summary, investigating the mechanisms of vascular calcification in ECs and VSMCs is crucial for understanding and treating this condition effectively. The use of advanced imaging techniques helps identify early signs of cardiovascular problems, leading to improved disease management and patient outcomes.

## Figures and Tables

**Figure 1 diagnostics-13-02632-f001:**
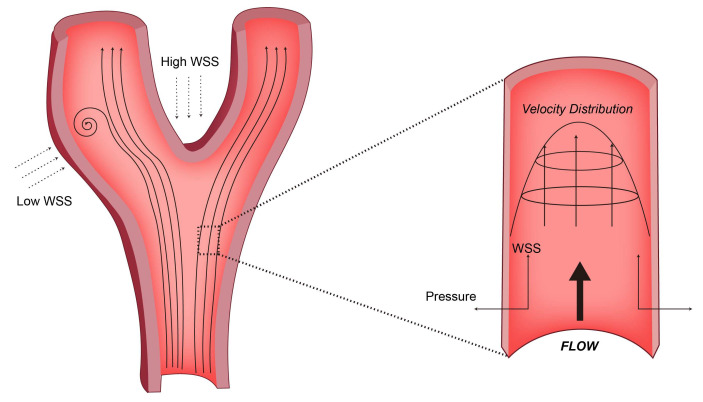
Schemes follow the same formatting. The straight segments of the arterial tree exhibit predominant laminar flow, while the presence of curvature and branch points (such as the aortic arch and iliac vessel bifurcation) leads to flow disturbances characterized by reduced WSS. For the force exerted by the blood flow on the vessel wall, the WSS is tangential to the vessel wall, while the pressure is perpendicular.

**Figure 2 diagnostics-13-02632-f002:**
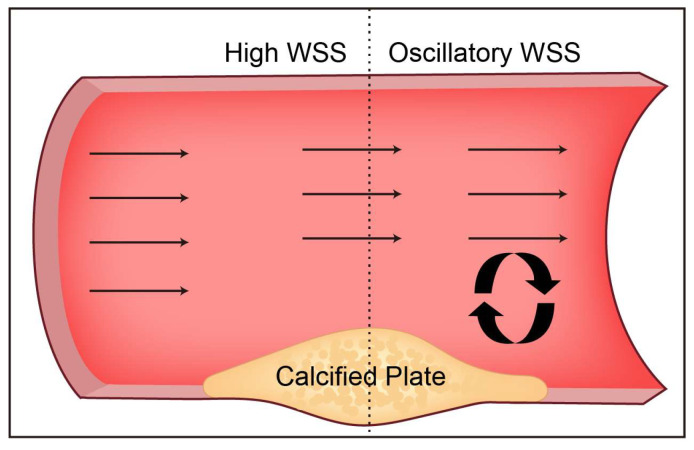
The distribution characteristics of WSS around the calcified plaque in the straight artery protruding from the lumen. Calcified plaque narrows the arterial lumen with high WSS in the upstream shoulder and disturbed flow downstream with low WSS.

**Figure 3 diagnostics-13-02632-f003:**
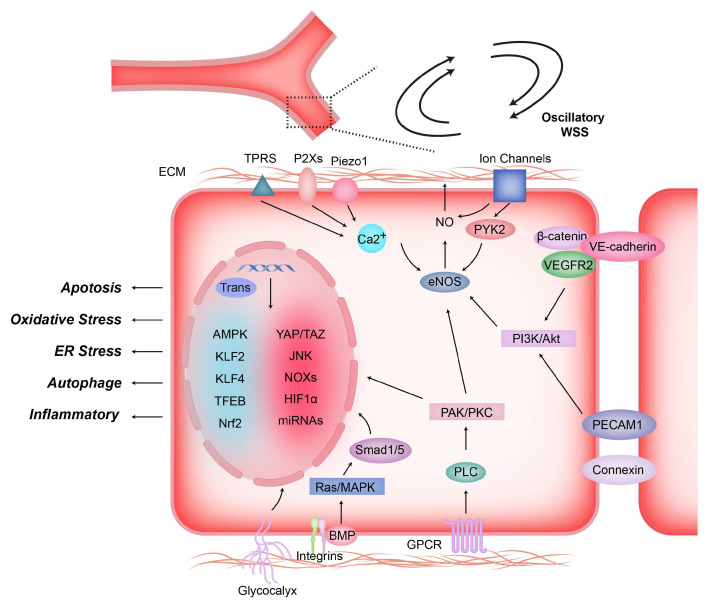
Shear stress induces mechanotransduction and signal transduction in endothelial cells. Shear-stress-induced mechanical forces stimulate the activation of mechanosensors located on the surface of endothelial cells, including G protein-coupled receptors (GPCRs), integrins, ion channels, glycocalyx, intercellular junction proteins (e.g., VE.cadherin and PECAM-1), non-selective Ca^2+^ channels, etc. The sensor activates signaling molecules inside ECs after sensing mechanical signals, such as PAK, PKC, PI3K, AKt, etc., which then regulate NO production, gene expression, and biological behavior of endothelial cells by activating eNOS, Smad1/5 and transcription factors.

**Figure 4 diagnostics-13-02632-f004:**
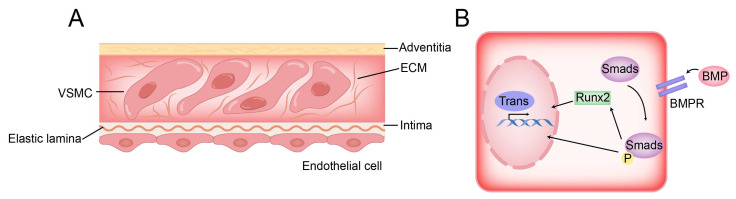
Structure of vessel wall and BMP pathway of VSMC osteogenic transformation. (**A**) VSMCs are distributed in the ECM and separated from endothelial cells by elastic lamina. (**B**) In the BMP pathway, BMP activates the osteogenic gene of VSMC directly or indirectly by activating Smad protein.

## Data Availability

Not applicable.

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
