# Peer review of "Mechanism Analysis of Vascular Calcification Based on Fluid Dynamics"

_diagnostics, 2023, doi:10.3390/diagnostics13162632_

Round 1

Reviewer 1 Report

The article describes the complex mechanism involved in calcification, from physical stimuli to heterocellular interactions. The promising imaging techniques can not capture these mechanisms yet, but the advances in artificial intelligence are speeding up the field.

I recommend the publication of this work.

Minor comments

Some abbreviations must be revised and defined as first mentioned; for example, fractional flow reserve (FFR) in line 483 was mentioned as FFR in lines 444  and 447

Reviewer 2 Report

I am interested in this article. The authors described the mechanism of vascular calcification by the molecular biological aspect and contributed the diagnostic strategy such as 4D floe MRI and computed fluid dynamics. This newly strategy will let the early diagnosis ability improve dramatically. I would like to request that the authors explain about 4D fluid MRI and demonstrate its image.

I have no idea, because I am not native English speaker.

Reviewer 3 Report

Comments and Suggestions for Authors

The manuscript is a narrative review that brings together information about the important aspects related to vascular calcification based on fluid dynamics, from pathogenesis to clinical implications and state-of-art imagistic techniques used for diagnostic and possible therapeutic strategies.

The manuscript is well written in terms of the English language, and it is also logically structured. The topic is of great interest for researchers, but also for clinicians. However, it requires improvement by reviewing a few issues:

Minor:

1.      Throughout the text, the authors use a colloquial expression, namely "and so on" (lines 38, 92, and 181). Please replace it.

2.      In chapter 3, the authors showed the difference between wall shear stress (WSS) in arteries and veins, noting that it is very low in veins compared to arteries, but then abandoning the approach of extremely low calcifications found in veins. Please add and expand the explanation of why calcifications are less frequent in veins than in arteries, taking into account the paradoxical fact that WSS is very low and also there is no stable WSS (which is protective against calcification), due to the existence of sense valves in veins.

3.      In the conclusion chapter, the authors introduced a phrase that was not addressed in the review in the context of the existing literature, namely:”The low WSS of disturbed flow increases the vascular 510 permeability of cholesterol-rich lipoproteins” (lines 510-511). In this sense, either this idea is discussed with rigorous references in the review, or it is removed from the conclusions.

Major:

1.      In chapter 2 (Overview of vascular calcification), the authors did not address at all one of the fundamental theories of vascular calcifications, namely the imbalance between pro- and anti-calcification factors/triggers. Please briefly address this idea as well.

2.      In chapter 4.2 (Osteogenic transformation of VSMCs during calcification), matrix Gla protein (MGP), one of the pivotal proteins in the fight against calcification secreted by VSMCs, was not mentioned at all, especially since the authors extensively addressed the idea of matrix stiffness in which MGP is directly involved. Please briefly discuss the role of this protein in this chapter.

Overall, I enjoyed reading this well-structured and clear review.

Minor editing of the English language is required.
